# Sociodemographic and Clinical Characteristics Associated with Suicidal Behaviour and Relationship with a Nurse-Led Suicide Prevention Programme

**DOI:** 10.3390/ijerph17238765

**Published:** 2020-11-25

**Authors:** Judit Pons-Baños, David Ballester-Ferrando, Lola Riesco-Miranda, Santiago Escoté-Llobet, Jordi Jiménez-Nuño, Concepció Fuentes-Pumarola, Montserrat Serra-Millàs

**Affiliations:** 1Department of Psychiatry and Mental Health, Consorci Hospitalari de Vic, 08500 Vic, Spain; judit.pons@uvic.cat (J.P.-B.); lola.riesco@uvic.cat (L.R.-M.); sescote@chv.cat (S.E.-L.); mserram@chv.cat (M.S.-M.); 2Faculty of Health Sciences and Wellbeing, University of Vic—Central University of Catalonia, 08500 Vic, Spain; 3Interinstitutional Research Group, Department of Mental Health and Social Innovation, 08500 Vic, Spain; 4Health and Healthcare Research Group, Department of Nursing, University of Girona, 17003 Girona, Spain; concepcio.fuentes@udg.edu; 5Vic Forensic Medicine Department, Institute of Legal Medicine and Forensic Sciences of Catalonia, 08500 Vic, Spain; jordi.jimenez@xij.gencat.cat

**Keywords:** preventive health services, psychiatric nursing, risk factors, suicidal ideation, suicide, attempted, suicide, completed

## Abstract

Suicidal behaviour is a major public health problem that needs to be tackled by all health agents including mental health nurses. Aims: The purpose of this study was to analyse the relationship between demographic and clinical characteristics and different kinds of suicidal behaviour with a nurse-led suicide prevention programme. Methods: The design was a cross-sectional study, performed in the region of Osona (Catalonia) in the five-year period 2013–2017. Suicidal behaviour was classified as suicidal ideation, interrupted self-directed violence, suicide attempt or completed suicide. Results: The sample included 753 patients (of whom 53 completed suicide) who experienced 931 suicidal behaviour episodes. Men represented only 38.4% of the sample but 81.1% of completed suicides. Mental disorders were associated with suicidal behaviour in 75.4% of the sample. Two thirds (66.4%) of the individuals (0.8% (*n* = 4) of whom completed suicide) were participants in a nurse-led suicidal behaviour case management programme. Conclusion: The main risk factors were being a woman for suicidal behaviour and being a man and being older for completed suicide. Mental disorders, widowhood and retirement were also associated with completed suicide. The completed suicide rate was lower among participants in the nurse-led programme.

## 1. Introduction

Suicidal behaviour is a public health problem that results in 800,000 deaths worldwide each year; in addition, for every adult who completes suicide, another 20 are estimated to have attempted suicide [1].

Mental health nurses may be instrumental in preventing suicidal acts by patients as they are in a position to recognize and respond to expressions of mental distress that are possible warning signs of suicidal behaviour [2]. Nurses can therefore play an essential role in suicide prevention programmes, especially in outpatient services, as relational factors underpin the communication of suicidal behaviour or intentions and the prevention of repeated suicide attempts [3].

### Background

Suicidal behaviour occurs along a continuum that reflects the severity of suicidal tendencies, ranging from fleeting and unplanned suicidal ideation to completed suicide. In the interest of advancing research worldwide, experts have pointed to the importance of reaching a consensus on definitions [4,5].

Certain demographic characteristics are associated with completed suicide. In terms of age, suicide rates are higher among middle-aged and elderly men, although rates are also increasing among teenagers in high-income countries [6]. In teenagers and young adults, the worldwide reported lifetime prevalence rates for suicidal ideation and suicidal behaviour are 12.1%–33% and 4.1%–9.3%, respectively [6]. Regarding sex, suicide rates worldwide are higher for men than for women, with some exceptions, e.g., China [6]. In Europe, while completed suicides by men outnumber suicides by women by 3.47, suicidal ideation and attempt rates are higher among women [7].

Risk factors for completed suicide include socioeconomic factors [8], with a number of studies pointing, in particular, to a significant association between completed suicide and unemployment [9]. Economic recessions have also been reported to have a negative impact on suicide rates [9,10,11,12]. Poor quality relationships and interpersonal conflicts are other contributing factors, as they play an important role in precipitating suicidal behaviour, while the loss of a partner through divorce/separation or widowhood is also associated with an increase in completed suicides [8,13]. With regard to the impact of migration, it has been reported that migrants from high to low suicide-risk countries are at a greater risk of completing suicide than the host population and vice versa [14].

Psychiatric comorbidity is one of the most important risk factors for completed suicide, most especially mood disorders, borderline personality disorder and substance use disorders [15]. Despite the fact that the prevalence of mental disorders in completed suicides is widely believed to be 90% [16], some studies have pointed to worldwide rates of 80%, with great variability depending on the geographical area [16,17].

Another major factor associated with completed suicide is previous attempts, as individuals with a history of suicide attempts have a five- to six-fold greater risk of trying again [18]. Studies have shown an estimated repeat-attempt incidence of 16.3% at one year, 16.8% at two years and 22.4% at five years, and of suicide of 1.6% at one year, 2.1% at two years, 3.9% at five years and 4.2% at ten years [19]. It is estimated that people who have made a previous suicide attempt are 66 times more likely to complete suicide than people with no such history [20].

Follow-up and community support have been effective in reducing suicide and suicide attempt rates among patients who have been recently discharged from hospital [21]. People who attempt suicide are recommended to be followed up using suicide prevention strategies, despite scarce and varying evidence on the chain of care and follow-up interventions [22]. The fact that the risk of completed suicide peaks immediately after hospital discharge following a suicide attempt only underscores the need for provision of early and effective follow-up care [23].

Community-based case management has been shown to be an effective intervention that reduces healthcare costs. Mental health nurse-led case management, in particular, controls patient symptoms and ultimately improves outcomes, cost effectiveness and patient satisfaction, even for individuals who have attempted suicide [24]. Mental health nurses have, in fact, been shown to assess suicide risk in a manner comparable to psychiatrists [25]. Follow-up through a nurse-led case management programme may therefore be beneficial for individuals with suicidal behaviour.

This study aimed to identify the sociodemographic and clinical characteristics of individuals with suicidal behaviour in the Osona region of Catalonia in terms of suicidal ideation, interrupted self-directed violence, attempted suicide and completed suicide, and to analyse what differences there were, if any, between non-participants and participants in a nurse-led prevention programme.

## 2. Materials and Methods

### 2.1. Sample and Data

The period of study was 2013–2017 and the study centre was the Vic Hospital Consortium (CHV), a public health consortium whose catchment area is the Osona region of Catalonia (population 156,572) [26].

The sample for this five-year cross-sectional descriptive study was composed of all completed suicides recorded for our region and individuals with suicidal behaviour (suicidal ideation, interrupted self-directed violence, attempted suicide) attended to in our hospital emergency room.

The data for all cases included in this study were collected by a research team nurse, who drew data from medical records for individual patients from the hospital registry and from a forensic registry. The hospital registry includes all suicide attempts and interrupted self-directed violence incidents attended to in the emergency room, and also any cases of suicidal ideation that the duty psychiatrist recommends for follow-up care. The forensic registry, created in 2006 by means of a collaboration agreement between CHV and the Institute of Legal Medicine and Forensic Sciences of Catalonia (Institut de Medicina Legal i Ciències Forenses de Catalunya-IMLCFC), includes data on completed suicides [27].

### 2.2. The Suicidal Behaviour Case Management Programme

The CHV Suicidal Behaviour Case Management Programme (SBCMP) is led by a mental health nurse, who provides care (face-to-face and by telephone) over 12 months to adult patients with suicidal behaviour referred by the emergency room. Patient participation is voluntary. In an initial face-to-face visit, held in the week after hospital discharge, the therapeutic relationship is initiated and health and other aspects related to the patient’s suicidal behaviour are evaluated, along with adherence to treatment, persistence of precipitating factors and current suicide risk.

Patients are provided with a direct-contact telephone number and an open schedule for consultations so that they can contact the nurse in person as often as necessary.

Visits with the patient cover aspects as follows [28]:Coping, i.e., assisting a patient to adapt to perceived stressors, changes or threats that interfere with their role in life;Crisis intervention, i.e., crisis counselling aimed at helping the patient cope with crises and resume a state of functioning comparable to or better than the pre-crisis state;Anxiety reduction, i.e., minimizing apprehension, dread, foreboding and uneasiness related to an unidentified source of anticipated danger;Counselling, i.e., interactively helping the patient by focusing on needs, problems or feelings (their own and of significant others) and enhancing or supporting problem solving and interpersonal relationships;Cognitive restructuring, i.e., challenging a patient to alter distorted thought patterns and view the self and the world more realistically;Preventing substance abuse (alcohol or drugs).

Scheduling of visits with other health professionals is also reviewed and new visits are scheduled as necessary and/or the patient is referred to other units. At one, three and six months after the suicidal behaviour that led to inclusion in the SBCMP, monitoring telephone calls are routinely made to check adherence to treatment and persistence of precipitating factors and to re-evaluate suicide risk.

The SBCMP concludes after 12 months with a face-to-face visit in which the usual aspects are evaluated. The episode is considered closed at this point, although the patient is encouraged to re-consult as needed using the usual channels and SBCMP channels (direct telephone and open consultation).

### 2.3. Variables

Four main categories of data were collected, namely, suicidal behaviour, sociodemographic data, clinical data and participation/non-participation in the SBCMP.

1. Suicidal behaviour was classified [29,30] as follows:Suicidal ideation: thinking about, considering, or planning suicide;Interrupted self-directed violence: taking steps to injure oneself but stopped by the self/another person prior to fatal injury;Suicide attempt: a non-fatal self-directed potentially injurious act performed with the intention to die, possibly but not necessarily resulting in injury;Completed suicide: self-inflicted death with evidence of an intention to die.

Note that profiles that reflected more than one episode of suicidal behaviour were classified according to the most serious episode, i.e., in the following order: completed suicide > suicide attempt > interrupted self-directed violence > suicidal ideation.

2. Sociodemographic data analysed were age, sex, country of birth, civil status, occupational status, family composition and rural/urban residence.

3. Clinical data collected were history, number and type of mental disorders, referrals to the mental health service in the previous year, suicidal behaviour in the previous year and presence of toxic substances in urine. Hospitalization details were also recorded, including length of stay and department.

4. SBCMP participation/non-participation, i.e., whether the included patient participated or not in the SBCMP.

### 2.4. Ethical Considerations

The research was performed according to the principles of the Declaration of Helsinki. The study was approved by the Clinical Research Ethics Committee of the Osona Foundation for Healthcare Research and Education (FORES) on 14 April 2016 (code 2016906/PR137).

### 2.5. Data Analysis

Qualitative variables were expressed as frequencies and percentages, and quantitative variables as means and standard deviations (SD). Contingency tables were created for the bivariate and multivariate analyses. Qualitative variables were compared using the chi square test and, when the expected count was less than five, Yates’ correction or Fisher’s exact test was used to readjust the variables. Quantitative and qualitative variables were compared using analysis of variance (ANOVA). Univariate and multivariate logistic regression analyses identified the variables most associated with completed suicides. The data were analysed using SPSS v.22 (IBM, Armonk, NY, USA), for a confidence interval (CI) of 95% and statistical significance of *p* ≤ 0.05.

## 3. Results

### 3.1. Characteristics of Suicidal Behaviour Episodes

Recorded during 2013–2017 were 931 suicidal behaviour episodes: 562 (60.4%) suicide attempts, 259 (27.8%) suicidal ideation episodes, 57 (6.1%) interrupted self-directed violence episodes, and 53 (5.7%) completed suicides. The mean (SD) episode rate was 119.79 (24.48) and the mean (SD) suicide rate was 6.82 (0.85) per 100,000 inhabitants/year [26].

The most frequently used methods were self-poisoning (83.1%; *n* = 467) for suicide attempts and asphyxia by hanging/strangulation/suffocation (43.4%; *n* = 23) for completed suicides (Table 1).

### 3.2. Sociodemographic Characteristics

Suicidal behaviour was manifested by 753 individuals, 464 (61.6%) of whom were women. While the mean (SD) age of the sample overall was 43.44 (16.72) years (median 45; range 12–95), the mean (SD) age of the sub-group of individuals who completed suicide was 57.89 (17.51) years (median 54; range 27–91).

Of the 753 individuals manifesting suicidal behaviour, 116 had experienced more than one episode and the mean (SD) number of episodes was 2.53 (1.1).

Immigrants represented 14.6% (*n* = 110) of the sample. By continent, nearly half were from America (42.7%; *n* = 47), over a third were from Africa (37.3%; *n* = 41) and the remainder were from other European countries (17.3%; *n* = 19) and Asia (2.7%; *n* = 3). By nationality, Moroccans accounted for a third of the immigrants (32.7%; *n* = 36), followed by Colombians (13.6%; *n* = 15) and Ecuadorians (8.2%; *n* = 9).

In terms of civil status at the time of the suicidal behaviour, 37.5% (*n* = 282) were in a stable relationship, 28.4% (*n* = 214) were separated/divorced, 26.2% (*n* = 197) were single and 6.2% (*n* = 47) were widowed.

Nearly half of the sample were employed (46%; *n* = 335) and the remainder were either unemployed (28.7%; *n* = 209) or retired (25.4%; *n* = 185). Around half of the patients lived with a formed family (52%; *n* = 383), just under a quarter lived with their family of origin (24%; *n* = 177) or alone (22.7%; *n* = 167) and 1.2% (*n* = 9) lived in a residence (Table 2).

Civil status analysed by type of suicidal behaviour revealed that widowed persons accounted for nearly a fifth of those who completed suicide (18.8%; *n* = 9), but for only a small proportion of cases of ideation (5.7%; *n* = 12), self-directed violence (4.4%; *n* = 2) and suicide attempts (5.5%; *n* = 24). However, Yates’ correction rendered this result non-significant.

### 3.3. Clinical Characteristics

Three quarters of the overall sample had a mental disorder (75.4%; *n* = 568), half had been seen by a mental health professional in the previous year (50.3%; *n* = 379) and 7.6% (*n* = 57) had experienced an episode of suicidal behaviour in the previous year (Table 3).

Hospitalization was necessary for a fifth of the episodes (21.1%; *n* = 196). Mean (SD) stay was 17.90 (25.91) days (median 9), with half of the patients admitted to an acute psychiatric unit (49%; *n* = 96), a quarter to an observation unit (26%; *n* = 51) and 11.7% (*n* = 23) to an intensive care unit (ICU). In 43.4% (*n* = 404) of the episodes, patients underwent urine testing for substance use, with nearly half (43.3%; *n* = 175) testing positive: 24.75% (*n* = 100) for alcohol, 13.12% (*n* = 53) for cocaine, 11.9% (*n* = 48) for cannabis and 4.2% (*n* = 17) for other substances.

### 3.4. Characteristics Related to the SBCMP

Two thirds of the sample (66.4%; *n* = 500) were included in the SBCMP at some point, mainly after the first recorded episode (93.6%; *n* = 468). Small numbers of patients were treated twice (6%; *n* = 30) and three times (0.4%; *n* = 2) in the SBCMP. Of the patients treated in the SBCMP, 0.8% (*n* = 4) completed suicide.

Of the patients who had experienced more than one episode (*n* = 116), the proportions who completed suicide were 4.1% (*n* = 4) for participants in the SBCMP and 5.3% (*n* = 1) for non-participants in the SBCMP (no significant difference) (Table 4).

### 3.5. Factors Associated with Completed Suicides

Logistic regression was based on comparing patients who died by suicide with the remaining patients. Univariate regression was performed with the statistically significant variables. A multivariate model based on forward selection (adding age, sex, civil status, occupational status, family composition, urban/rural residence and mental disorder variables) pointed to associations between suicide and being male, being older and a lower probability of having a mental disorder (Table 5).

## 4. Discussion

Our study yields new data in relation to research into suicidal behaviour, with some of our results confirming, and others diverging from, the findings of previous studies.

Regarding suicide methods, our results corroborate other studies that have reported that methods used for suicide attempts are less potentially lethal than for completed suicides. The most frequently used method for suicide attempts by our patients was self-poisoning (83.10%) followed by cutting (10.5%), corroborating findings elsewhere regarding self-poisoning (82.4%) and cutting (7.1%) [31]. The most frequently used methods for completed suicides were asphyxia (43.4%) and jumping from tall places (17%), coherent with national data for Spain [32]. However, our results diverge from Spanish data thereafter, as suicide by jumping/lying before a moving object was the third most frequent method (15.10%) in our study, but ninth in Spain, whereas suicide by self-poisoning occupied third place for Spain [32]. One possible explanation for this difference is that the study region (Osona) has a railway line but no underground rail system.

In terms of age, our results largely corroborate the literature for high-income countries [6,33], which points to a statistically significant difference of some 17 years in the mean age of people who complete suicide compared to those who attempt suicide. In our study, the median age of those who completed suicide was 54 years, higher than the 49.5 years reported elsewhere [34]. In our regression, the odds ratio (OR) for age was 1.06 (1.033–1.078), indicating that the suicide risk increased with each additional year (note that there were no completed suicides by teenagers in our sample).

Regarding sex, our findings confirm the “sex paradox” reported in previous studies [33]; thus, while women accounted for most suicide attempts (70.6%) in our sample, men accounted for most completed suicides (81.1%); moreover, our suicide rate for men was much higher than the 74.6% reported for Spain [35]. Our results—except for interrupted self-directed violence, where we found men to be in the majority (53.3%)—corroborate the results of a review for developed countries [7] that found that women were in the majority in all types of suicidal behaviour except for completed suicide. Future studies with a larger sample would enable us to determine whether cases of interrupted self-directed violence are more similar to cases of completed suicide than to cases of suicide attempts and suicidal ideation.

In relation to immigrants, our study confirms findings reported elsewhere that suicide rates among immigrants are no higher than for the general population [14]. Immigrants in our study accounted for 14.6% of suicidal behaviour overall, a proportion very similar to the 13.35% (0.54%) mean (SD) immigrant population in the catchment area [36]; the percentage of completed suicides by immigrants was lower (7.5%), whereas the percentage of suicide attempts was higher (17.9%). Suicide rates for immigrants from Morocco, Colombia and Ecuador were similar to rates in the countries of origin (crude all-age suicide rates per 100,000 population of below 5.0 for Morocco and 5.0–9.9 for Colombia and Ecuador [1,21].

In relation to civil status, the fact that 18.8% of patients who completed suicide were widowed corroborates the literature. However, we found no association between separation/divorce and completed suicide [13,37], with ORs of 0.44 (0.18–1.07) and 0.85 (0.41–1.78) for being separated/divorced and single, respectively, relative to being in a stable relationship. These results contradict those of two reviews, one that pointed to a negative association between being in a stable relationship and completed suicide [8] and another that reported, for singles, an increased risk of both completed suicide and suicidal ideation but not of suicidal intention [38].

We found statistically significant differences regarding occupational status, with retirement being most frequently associated with completed suicide (53.2%). This result is coherent with our other finding regarding suicide completion being more frequent in older age groups. Employed people accounted for 31.9% of deaths by suicide in our study, a higher percentage than the 22.9% reported by Gómez-Durán and colleagues [34], although that latter finding only referred to employed patients with a mental health history. The fact that we found no association between unemployment and completed suicide contradicts Milner and colleagues [9], who reported a relative risk (RR) of 1.41 (1.21–1.60) for unemployment, whereas our finding was an OR of 0.74 (0.29–1.84) for unemployment. Note that we did not analyse the quality of employment, which has recently been reported as a factor for consideration [39].

Our patients, irrespective of type of suicidal behaviour, mostly lived with family. Our finding of 40.9% of patients living alone or in a residence who completed suicide—OR = 1.87 (0.98–3.58)—was noticeably higher than the 28.6% reported by Gómez-Duran and colleagues [34]. Note that we only considered living arrangements and not feelings of loneliness, as discussed elsewhere [40].

While we found an association between mental disorders and completed suicide in our patients, this association differed from that reported in the literature. In most other studies, while mental disorders are reported to be associated with 80%–90% of completed suicides [6,16,17] and are also associated with suicide attempts [41], in our sample, and contrasting with an overall rate for our sample of 75.4%, only 56.6% of our patients who completed suicide had mental disorders. Gómez-Durán and colleagues [34] found that only 45.5% of those who completed suicide were diagnosed with a mental disorder; however, unlike our researchers, they had access to very few medical records. The fact that a mental disorder may be incipient or undiagnosed would support the importance of early detection in primary healthcare as a suicide prevention measure [21]. Future studies should monitor whether this pattern persists, although it needs to be kept in mind that findings may be influenced by suicide prevention measures already being implemented in our catchment area. Our study points to lower psychiatric comorbidity for patients who completed suicide (adjusted OR = 2.37: 1.18–4.76), even though psychiatric comorbidity was higher among the other patients in our sample, a finding that corroborates the findings of the review by Cho and colleagues [17].

The main mental disorders diagnosed among the patients in our sample who completed suicide were mood disorders and disorders associated with substance use (24.5% each). This finding corroborates a meta-analysis [42], which reported ORs for Europe of 10.62 (4.50–25.09) and 6.54 (3.76–11.39) for suicide risk associated with those two disorders, respectively. Our percentage of mood disorders associated with completed suicide (24.5%) is lower than the 54.3% reported by Gómez-Durán and colleagues [34]. As for personality and psychotic disorders, Gómez-Durán and colleagues [34] reported a prevalence of 11.4% and 17.1%, respectively, contrasting with our prevalence of 7.5% each. Anxiety/adaptive disorders were present in just 9.4% of our patients who completed suicide (all five with adaptive disorders), contrasting with findings elsewhere [43] that post-traumatic stress disorder is the anxiety disorder most associated with completed suicide. Regarding other types of suicidal behaviour in our patients, mood disorders were most frequently associated with suicidal ideation (43.1%) and suicide attempts (30.3%), while anxiety/adaptive disorders were most frequently associated with interrupted self-directed violence (28.9%). In our sample, mental disorders related to substance use were associated with 24.5% of completed suicides and 16.4% of suicide attempts, a finding that diverges from reviews reporting a more important association for alcohol abuse with suicide attempts (OR = 3.13: 2.45–3.81) than with completed suicide (OR = 2.59: 1.95–3.23) or suicidal ideation (OR = 1.86: 1.38–2.35) [44,45].

Regarding previous suicidal behaviour, only 6% of our patients who completed suicide had had an episode in the previous year; this was a much lower percentage than the 33.3% reported by Bostwick and colleagues [46] or the 19.67% suicide attempt rate for the year prior to death reported by Mallon and colleagues [47].

Our study shows a high level of engagement by patients after discharge, as over two thirds (66.4%) voluntarily participated in the SBCMP. Even though the SBCMP targets patients aged 18 and older living in the hospital catchment area, included in our study were all individuals attending the emergency room for a suicidal behaviour episode, irrespective of age or area of residence. Our participation rate is substantially higher than rates reported elsewhere. Hunter [48] reported, for a large cohort study, that only 31.4% of patients attended a mental health visit within 30 days after emergency room treatment for deliberate self-harm, while Costemale-Lacoste [49], who analysed specialist out-treatment engagement of patients following suicide attempts, found that only 35% attended at least one visit in the first month after discharge; note however, that their focus was only patients who had not previously received psychiatric care.

The sociodemographic characteristics of patients participating in the SBCMP were significantly different from those who did not participate in terms of sex, civil status and occupational status, as more women, more divorced individuals and more unemployed individuals, respectively, participated. Sex-related differences can be explained by the fact that the suicide rate was higher among patients not participating in the SBCMP, and, as indicated by the review by Fox [7], completed suicide is more common among men, whereas suicide attempt and suicidal ideation is more common among women. Another explanation is that the mental health-related stigma regarding help-seeking is more prevalent in men than women [50,51].

There were also clinical differences between participants and non-participants in the SBCMP, in that participants had more mental disorders and more recorded episodes of suicidal behaviour. The higher proportion of patients with more than one suicidal behaviour episode (19.4% SBCMP vs 7.5% non-SBCMP) is explained by the SBCMP inclusion of patients with all type of episodes, including suicidal thoughts. Close contact is maintained with patients in the SBCMP, so if the nurse detects risk, the patient is encouraged to go to the emergency room for a psychiatric assessment. Our five-year rate of patients with more than one recorded episode (i.e., 19.4%) is nonetheless broadly similar to widely cited rates of 16% and 23% at one year and four years, respectively [52], and to the five-year 20.1% and two-year 18.9% reported by Parra-Uribe [53] and Irigoyen [54], respectively, for Spain.

Finally, the suicide rate (0.8%) for patients participating in our SBCMP, which correlated with higher psychiatric morbidity, was lower than the 2.51% and 0.9% of suicides completed one year after a suicide attempt reported for followed-up outpatients [55] and the 1.2% reported for a one-year telephone-based management programme [53].

However, it should be noted that, during the study period, most non-participants in the SBCMP who completed suicide had not been attended to for a previous suicidal behaviour episode. It is therefore important to consider establishing early detection mechanisms for such cases, focused on primary healthcare and mental health nurses as facilitating agents.

The findings of this study have to be seen in light of some limitations. While the forensic registry of completed suicides is reliable [27], because the real number of deaths is small, we cannot establish robust associations. Additionally, our sample should be considered incomplete, because the hospital registry only includes suicidal ideation episodes treated in the emergency room if a psychiatrist specifically refers the patient for follow-up; this explains why our study reflects fewer ideation episodes than suicide attempts. With regard to previous suicide attempts, our study only reflected attempts in the previous year whereas future studies should take into account all previous suicide attempts by patients. Regarding relationships and employment, the fact that the quality of either was not taken into account could have affected our findings. Finally, it should also be kept in mind that our comparisons were drawn between people with suicidal behaviour and not with the general population.

## 5. Conclusions

Age was a key factor in completed suicides in our sample, with higher suicide mortality in older age groups. Sex clearly determined forms of suicidal behaviour: while 8 in 10 completed suicides were by men, 7 in 10 suicide attempts were by women. Coherent with the finding of higher suicide mortality among older people was the fact that the risk of completed suicide was 3.3 times higher for retired individuals than for employed individuals and 3.1 times higher for widowed individuals than for people with partners. Mental disorders were less frequent in patients who completed suicide than in patients with other suicidal behaviours. The most prevalent mental disorders in our sample were mood disorders. While a high proportion of patients with mental disorders participated in the SBCMP, the suicide completion rate for SBCMP patients was low comparative to other reported rates. Future research would require a larger sample size to confirm our findings and to explore health system instruments and programmes that could help prevent suicide.

## Figures and Tables

**Table 1 ijerph-17-08765-t001:** Completed suicide and attempted suicide methods.

	Attempted Suicide *n* (%) 562 (100)	Completed Suicide *n* (%) 53 (100)
Asphyxia (hanging/strangulation/suffocation)	7 (1.2)	23 (43.4)
Self-poisoning	467 (83.1)	2 (3.8)
Cutting/stabbing	59 (10.5)	2 (3.8)
Jumping from a tall place	9 (1.6)	9 (17)
Jumping/lying before a moving object	3 (0.5)	8 (15.1)
Firearms/airguns/explosives	1 (0.2)	4 (7.5)
Drowning	2 (0.4)	3 (5.7)
Other means	14 (2.5)	2 (3.8)

**Table 2 ijerph-17-08765-t002:** Sociodemographic characteristics according to type of suicidal behaviour.

		Suicidal Ideation *n* (%) 209 (100)	Interrupted Self-Directed Violence *n* (%) 45 (100)	Suicide Attempt *n* (%) 446 (100)	Completed Suicide *n* (%) 53 (100)	Test	*p* ^†^
Age, mean (SD)		45.62 (14.4)	44.78 (16.09)	40.57 (16.69)	57.89 (17.51)	F = 20.29	<0.001 ^a^
Sex	Man	91 (43.5)	24 (53.3)	131 (29.4)	43 (81.1)	F = 22.74	<0.001 ^b^
	Woman	118 (56.5)	21 (46.7)	315 (70.6)	10 (18.9)		
Immigrant	No	188 (90)	40 (88.9)	366 (82.1)	49 (92.5)	F = 3.36	0.018 ^c^
	Yes	21 (10)	5 (11.1)	80 (17.9)	4 (7.5)		
Stable relationship	Yes	71 (34)	19 (42.2)	172 (39.3)	20 (41.7)	F = 0.78	0.505
	No	138 (66)	26 (57.8)	266 (60.7)	28 (58.3)		
Occupational status	Employed	99 (47.4)	21 (46.7)	200 (46.7)	15 (31.9)	F = 4.5	0.004 ^d^
	Unemployed	63 (30.1)	15 (33.3)	124 (29)	7 (14.9)		
	Retired	47 (22.5)	9 (20)	104 (24.3)	25 (53.2)		
Family composition	Formed family	108 (51.7)	24 (53.3)	229 (52.3)	22 (50)	F = 1.73	0.160
	Family of origin	38 (18.2)	12 (26.7)	123 (28.1)	4 (9.1)		
	Living alone/in residence	63 (30.1)	9 (20)	86 (19.6)	18 (40.9)		
Urban/rural	Rural	107 (52.2)	17 (39.5)	195 (45.2)	27 (52.9)	F = 1.46	0.22
	Urban	98 (47.8)	26 (60.5)	236 (54.8)	24 (47.1)		

^†^ Tukey post-hoc test differences: ^a^ suicide attempt vs suicidal ideation (*p* = 0.001), completed suicide vs suicidal ideation (*p* < 0.001), completed suicide vs interrupted self-directed violence (*p* < 0.001), completed suicide vs suicide attempt (*p* < 0.001). ^b^ Suicide attempt vs suicidal ideation (*p* = 0.002), suicide attempt vs interrupted self-directed violence (*p* = 0.006), suicide attempt vs completed suicide (*p* < 0.001), completed suicide vs suicidal ideation (*p* < 0.001), completed suicide vs interrupted self-directed violence (*p* = 0.018). ^c^ Suicide attempt vs suicidal ideation (*p* = 0.038). ^d^ Completed suicide vs suicidal ideation (*p* = 0.003), completed suicide vs interrupted self-directed violence (*p* = 0.025), completed suicide vs suicide attempt (*p* = 0.003).

**Table 3 ijerph-17-08765-t003:** Clinical characteristics according to type of suicidal behaviour.

	Suicidal Ideation *n* = 209 (100%)	Interrupted Self-Directed Violence *n* = 45 (100%)	Suicide Attempt *n* = 446 (100%)	Completed Suicide *n* = 53 (100%)	Test	*p* ^†^
Previous mental disorder, *n* (%)	164 (78.5)	29 (64.4)	345 (77.4)	30 (56.6)	F = 5.07	0.002 ^a^
Previous diagnoses, mean (SD)	1.30 (0.498)	1.34 (0.484)	1.24 (0.461)	1.37 (0.615)	F = 1.26	0.288
Mental health referral in the previous year, *n* (%)	117 (56)	17 (37.8)	223 (50)	22 (41.5)	F=2.4	0.067
Suicidal behaviour in the previous year, *n* (%)	7 (3.3)	1 (2.2)	46 (10.4)	3 (5.9)	F = 4.13	0.006 ^b^
DISORDERS, *n* (%) ^‡^						
Mood	90 (43.1)	11 (24.4)	135 (30.3)	13 (24.5)	F = 4.82	0.002 ^b^
Psychotic	2 (1)	1 (2.2)	16 (3.6)	4 (7.5)	F = 2.43	0.064
Anxiety/adaptive	54 (25.8)	13 (28.9)	114 (25.6)	5 (9.4)	F = 2.47	0.06 ^c^
Personality	27 (12.9)	5 (11.1)	51 (11.4)	4 (7.5)	F = 0.41	0.747
Substance use-related	31 (14.8)	7 (15.6)	73 (16.4)	13 (24.5)	F = 0.97	0.402
Other	9 (4.3)	2 (4.4)	39 (8.7)	2 (3.8)	F = 1.93	0.123

^†^ Tukey post-hoc test differences: ^a^ completed suicide vs suicidal ideation (*p* = 0.005), completed suicide vs suicide attempt (*p* = 0.005). ^b^ Suicide attempt vs suicidal ideation (*p* = 0.009). ^c^ Completed suicide vs suicide attempt (*p* = 0.049). ^‡^ Each patient may have more than one disorder.

**Table 4 ijerph-17-08765-t004:** Sociodemographic and clinical characteristics according to inclusion in the Suicidal Behaviour Case Management Programme (SBCMP).

		Not Included in SBCMP *n* (%) 253 (100)	Included in SBCMP *n* (%) 500 (100)	Test	*p*
Age, mean (SD)		42.9 (20.7)	43.72 (14.3)	F = 55.82	0.572
Sex	Man	120 (47.4)	169 (33.8)	F = 13.2	<0.001
	Woman	133 (52.6)	331 (66.2)		
Civil status	Single	94 (39.2)	103 (20.6)	F = 34.86	<0.001
	Married	82 (34.2)	200 (40)		
	Divorced	46 (19.2)	168 (33.6)		
	Widowed	18 (7.5)	29 (5.8)		
Occupational status	Employed	118 (51.1)	217 (43.6)	F = 24.23	<0.001
	Unemployed	39 (16.9)	170 (34.1)		
	Retired	74 (32)	111 (22.3)		
Mental disorder	No	77 (30.4)	108 (21.6)	F = 7.08	0.005
	Yes	176 (69.6)	392 (78.4)		
Number of mental disorders, mean (SD)		1.21 (0.44)	1.3 (0.5)	F = 14.95	0.039
Recorded episodes	1	234 (92.5)	403 (80.6)	F = 18.22	<0.001
	>1	19 (7.5)	97 (19.4)		

**Table 5 ijerph-17-08765-t005:** Raw and adjusted odds ratio (OR) values for factors associated with completed suicide.

		OR (CI 95%) Raw	OR (CI 95%) Adjusted
Age		1.057 (1.038–1.075)	1.055 (1.033–1.078)
Sex	Woman Man	1 7.936 (3.920–16.067)	1 6.798 (3.008–15.363)
Civil status	Stable relationship Separated/divorced Single Widowed	1 0.443 (0.184–1.068) 0.850 (0.405–1.781) 3.103 (1.31–7.311)	
Occupational status	Employed Unemployed Retired	1 0.739 (0.296–1.844) 3.333 (1.710–6.499)	
Family composition	Formed family Family of origin Living alone/in residence	1 0.379 (0.129–1.188) 1.869 (0.976–3.582)	
Rural/urban residence	Rural Urban	1 1.270 (0.72–2.25)	
Mental disorder	Yes No	1 2.546 (1.439–4.506)	1 2.375 (1.185–4.759)

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
