# Peer review of "Sociodemographic and Clinical Characteristics Associated with Suicidal Behaviour and Relationship with a Nurse-Led Suicide Prevention Programme"

_ijerph, 2020, doi:10.3390/ijerph17238765_

Round 1

Reviewer 1 Report

The manuscript addresses a relevant topic such as suicidal behaviors in people who have previously exhibited suicidal behaviors.

There are several inaccuracies, as well as inappropriate use of certain terms, for example, females instead of women, gender instead of sex, died by suicide instead of completed suicide, suicide rates or suicide deaths instead of suicide, and so on.

In the abstract, the instructions for the abstract preparation were included in the abstract itself. The background was missing.

There are several keywords and some of them are inaccurate. Modify and reduce the number of keywords.

In the background section, please, state that the suicide rates and sociodemographic characteristics reported are based on research in high-income countries, as Turecki & Brendt stated clearly. The last paragraph report twice the aim, choose only one of them.

In the Method’s section, regarding the participants, Were the patients engaged in other mental health services, for instance, psychiatric treatment, psychotherapy, support group, etc.? Do you have data on the patients' attachment to these services?  Report how and by who were collected the data. It seems that the SBCMP is just a follow-up program to assess suicide behavior, describe more the mental care aspect included in the program such as crisis intervention, psychiatric treatment, etc.

In the discussion, add a paragraph regarding sociodemographic and clinical characteristics integrated of the persons who died by suicide. Consider a more extensive discussion regarding gender aspects of suicide risk and the program’s attachment. Add more limitations to the study.

You will find more comments in the attachment.

Best regards

Reviewer 2 Report

This study is to analyze the characteristics of suicide-related events visited in the emergency room in catalonia and to show the effect of a nurse-led suicide prevention program.

This study should address several severe limitations.

1) It is unclear whether the hypothesis of the study is to analyze the characteristics and risk factors of suicide-related events admitted to the emergency room or to show the effect of a nursing-led suicide prevention program. If they were both, an irony of insufficient study occurred while trying to show both.

2) The authors' understanding of the epidemiological methodology is insufficient. Although this study is difficult to see causality, it expresses causality, and it is difficult to determine in what form the effect of the nurse-led program was measured.

3) Also, the less common narrative is making reviews more difficult. Authors should follow the general journal article's flow.

4) The methodology part is particularly difficult to understand.

5) I understand the value of the subject, but discoveries are already well known, and something new are needed.

Round 2

Reviewer 1 Report

The authors have addressed most of the recommendations made.

The manuscript has gained much in conceptual and methodological clarity, as well as in precision.

Only minor errors were identified in table 4 and reference 48 is missing in the reference list.

Reviewer 2 Report

Most of the issues I raised seem to have been resolved. Thank you for the hard work of the authors.